# Efficacy and Tolerability of Perampanel in Brain Tumor-Related Epilepsy: A Systematic Review

**DOI:** 10.3390/biomedicines11030651

**Published:** 2023-02-21

**Authors:** Jessica Rossi, Francesco Cavallieri, Maria Chiara Bassi, Giuseppe Biagini, Romana Rizzi, Marco Russo, Massimo Bondavalli, Corrado Iaccarino, Giacomo Pavesi, Salvatore Cozzi, Lucia Giaccherini, Masoumeh Najafi, Anna Pisanello, Franco Valzania

**Affiliations:** 1Clinical and Experimental Medicine PhD Program, University of Modena and Reggio Emilia, 41125 Modena, Italy; 2Neurology Unit, Neuromotor & Rehabilitation Department, Azienda USL-IRCCS of Reggio Emilia, 42123 Reggio Emilia, Italy; 3Medical Library, Azienda USL-IRCCS of Reggio Emilia, 42123 Reggio Emilia, Italy; 4Department of Biomedical, Metabolic and Neural Sciences, University of Modena and Reggio Emilia, 41125 Modena, Italy; 5Neurosurgery Unit, Neuromotor and Rehabilitation Department, Azienda USL-IRCCS of Reggio Emilia, 42123 Reggio Emilia, Italy; 6Radiation Oncology Unit, Oncological Department and Advanced Technologies, Azienda USL-IRCCS of Reggio Emilia, 42123 Reggio Emilia, Italy; 7Radiation Oncology Department, Centre Léon Bérard, 69373 Lyon, France; 8Department of Radiation Oncology, Shohadaye Haft-e-Tir Hospital, Iran University of Medical Science, Teheran 1997667665, Iran

**Keywords:** brain, epilepsy, glioma, glutamate, perampanel, survival, tumor

## Abstract

(1) Background: Epilepsy is a frequent comorbidity in patients with brain tumors, in whom seizures are often drug-resistant. Current evidence suggests that excess of glutamatergic activity in the tumor microenvironment may favor epileptogenesis, but also tumor growth and invasiveness. The selective non-competitive α-amino-3-hydroxy-5-methyl-4-isoxazolepropionic acid (AMPA) receptor antagonist perampanel (PER) was demonstrated to be efficacious and well-tolerated in patients with focal seizures. Moreover, preclinical in vitro studies suggested a potential anti-tumor activity of this drug. In this systematic review, the clinical evidence on the efficacy and tolerability of PER in brain tumor-related epilepsy (BTRE) is summarized. (2) Methods: Five databases and two clinical trial registries were searched from inception to December 2022. (3) Results: Seven studies and six clinical trials were included. Sample size ranged from 8 to 36 patients, who received add-on PER (mean dosage from 4 to 7 mg/day) for BTRE. After a 6–12 month follow-up, the responder rate (% of patients achieving seizure freedom or reduction ≥ 50% of seizure frequency) ranged from 75% to 95%, with a seizure freedom rate of up to 94%. Regarding tolerability, 11–52% of patients experienced non-severe adverse effects (most frequent: dizziness, vertigo, anxiety, irritability). The retention rate ranged from 56% to 83%. However, only up to 12.5% of patients discontinued the drug because of the adverse events. (4) Conclusions: PER seems to be efficacious, safe, and well-tolerated in patients with BTRE. Further randomized studies should be conducted in more homogeneous and larger populations, also evaluating the effect of PER on tumor progression, overall survival, and progression-free survival.

## 1. Introduction

Seizures are one of the most frequent clinical manifestations in patients with brain tumors and represent the first symptom in 20–40% of patients [1]. Brain tumor-related epilepsy (BTRE) is drug resistant in 40% of cases [2], which contributes to a deterioration in the quality of life of patients, who are often forced to take anti-epileptic polytherapy. Adverse effects of antiepileptic drugs (AEDs) and interactions with chemotherapy are an important issue in BTRE treatment. First-generation of AEDs were characterized by significant enzyme-inducing or inhibiting properties and lower tolerability compared to the second-generation. Within the second-generation of antiepileptic drugs, levetiracetam (LEV) has been shown to be effective and tolerated in patients with brain tumors, and a recent systematic review demonstrated a 6-month seizure freedom rate of 39–96%, with a 6-month failure rate due to adverse effects and ineffectiveness of 1% and 10%, respectively [3]. However, data on a possible survival advantage of LEV in patients with brain tumors, in particular glioblastoma patients, are conflicting and a recent systematic review performed by Chen et al. concluded that LEV does not significantly improve survival in all patients with GBM [4]. An increasing understanding of the mechanisms of epileptogenesis in brain tumor patients and the advent of third-generation drugs have led to the need for targeted therapies in the treatment of BTRE.

A growing body of evidence suggests that the pathogenesis of BTRE is predominantly related to an excess of glutamatergic activity in the tumor microenvironment, resulting from excessive glutamate secretion by tumor cells and reduced glutamate clearance as well as from an increased expression of ionotropic α-amino-3-hydroxy-5-methyl-4-isoxazolepropionic acid (AMPA) and N-methyl-D-aspartate (NMDA) receptors at the level of tumor cells and peritumoral astrocytes [5]. Moreover, excessive glutamatergic tone might also favor tumor growth and invasiveness as well as excitotoxicity, an effect that leads to neurodegeneration and cognitive deterioration [5]. The role of glutamate in brain tumor-related epileptogenesis is widely known for gliomas [6], but have been established in other histotypes including meningiomas [7] and brain metastases [8].

Perampanel (PER), a relatively new selective non-competitive AMPA antagonist, has been initially authorized as an add-on treatment for patients with focal and focal to bilateral seizures, and more recently, for generalized onset seizures and as monotherapy in some countries [9]. Several randomized controlled trials [10,11] and metanalyses [12] proved that PER is efficacious and well-tolerated in patients with focal seizures. In addition, preclinical in vitro studies have demonstrated potential anti-tumor activity of PER [13], but these data have not been confirmed in rat models [14]. Currently, clinical data on the efficacy and safety of PER in BTRE as well as data on its antitumor effect are scarce.

This systematic review aims to summarize the clinical evidence about the efficacy and tolerability of PER in BTRE as well as the effect of PER on cognition and the natural history of brain tumors.

## 2. Methods

### 2.1. Primary and Secondary Objectives

We performed a systematic review of the evidence on the efficacy, safety, and tolerability of PER in patients with brain tumors from the currently available literature. The primary endpoint of this systematic review was to define the efficacy of PER in terms of the responder rate (seizure freedom or reduction ≥ 50% of seizure frequency compared to baseline assessment). The secondary endpoint was to assess the proportion of patients with adverse events after initiating PER as well as the type of adverse events and retention rate. When possible, we aimed to provide data about the effect of PER on cognition and on the natural history of the tumor. We also aimed to provide a comprehensive overview on the state-of-the-art from registered clinical trials evaluating the efficacy and safety of PER in BTRE. Recommendations of the Preferred Reporting Items for Systematic Reviews and Meta-Analyses (PRISMA) statement were followed while reporting this systematic review. No review protocol was previously registered.

### 2.2. Search Strategy

A systematic search was independently performed by M.B. and J. R. for all articles published until 13 December 2022 on “MEDLINE, EMBASE, SCOPUS, WEB OF SCIENCE, COCHRANE, CLINICALTIRAL.GOV, WHO TRIAL REGISTRY”.

We used the following keywords: “Glioma”, “brain tumor”, “brain tumour”, “brain cancer”, “brain neoplasm”, “brain metastas*”, “meningioma” Perampanel”, “Seizures”, “seizure”, and “epilepsy” (Appendix A).

### 2.3. Study Selection and Data Extraction

A first screening of results was performed by titles and abstracts. Prospective cohort studies, retrospective studies, and case series including more than five patients were included. We did not include single case reports or conference abstracts. The search was then restricted to studies with full text available in the English language, while studies in other than the English languages were excluded. Two authors (F.C. and J.R.) performed the selection and review of the articles. Following the identification of relevant studies, information from each article was independently extracted. Any disagreement between the two investigators on the inclusion of a study was solved by involving a third investigator (F.V.) in the discussion to reach a consensus decision.

The relevant data that were extracted after a full-text review of the included articles included the study design; initial sample size; the number of patients at the end of follow-up period; demographic characteristics of patients (sex and age); histology of the tumor, isocitrate dehydrogenase (IDH) and O^6^-methylguanine DNA methyltransferase (mgmt) gene methylation status; the proportion of patients undergoing to gross total resection (GTR); the proportion of patients treated with chemotherapy and/or radiotherapy; seizure semiology (focal, focal impaired awareness, focal to bilateral tonic-clonic seizures); the number of patients in monotherapy and polytherapy (taking two or more antiepileptic drugs) before adding PER; PER dosing regimen; follow-up duration; the proportion of patients with ≥50% and <50% reduction in seizure frequency; the proportion of patients with unchanged seizure frequency, complete seizure freedom, or worsening of seizures; type, frequency, and severity of adverse effects; the number of patients who prematurely discontinued PER due to adverse effect/ineffectiveness; and the retention rate. When available, we also extracted data regarding the proportion of patients experiencing improvement, stability, or worsening of cognitive function as well as the data on the effect of PER on tumor growth.

## 3. Results

### 3.1. Results of the Search

After a primary search, a total of 244 publications were screened. Among these, 108 were duplicates and removed. The eligibility of the remaining 136 papers was evaluated, and 103 irrelevant articles were excluded according to the title, article type, and abstract. Therefore, 33 articles underwent full text review. Out of these, three were narrative reviews; two studies did not provide information about epilepsy or the natural history of the tumor, and one addressed 22 patients with focal epilepsy, among which five had brain tumors (but histology nor other information about tumors were specified); one addressed five patients, among which two presented SMART syndromes as a confounding factor; one study was written in Russian, and one did not meet our primary or secondary endpoint. Furthermore, we did not include 11 conference abstracts (Figure 1).

Seven studies fulfilled the inclusion criteria and were included in the review (Figure 1). Six clinical trials were also separately analyzed.

### 3.2. Qualitative Review

#### 3.2.1. Studies on Brain Tumor-Related Epilepsy (BTRE)

Table 1 and Table 2 summarize the demographic and clinical data of patients included in the selected studies as well as the results regarding the effect of PER on BTRE.

Coppola et al. [2], in a prospective study, recruited 36 adult patients (males n = 23; females n = 13) aged between 15 and 75 years affected by BTRE: 11 patients with low grade glioma (LGG), 14 patients with high grade glioma (HGG), seven with glioblastoma, four unclassified. PER was added-on, and titrated from 2 mg/day up to a maximum of 12 mg/day. The mean dosage was 6.5 mg. Nineteen patients (52.8%) were on antiepileptic drug (AED) monotherapy with two or more antiepileptic drugs and 17 (47.2%) were on polytherapy before PER administration. Responder rate at 12 months in 21 patients was 90.4%: seven patients were seizure-free (33.3%), 12 had a seizure reduction ≥50% (57.1%), one remained stable, and one had a reduction ≤50%. Responder rate at the last follow-up available in the whole population (36 patients) was 66.6%: nine patients were seizure-free (25%), 16 had a seizure reduction ≥50% (41.6%), five remained stable, five had a reduction <50%, and two worsened. They observed a statistically significant mean seizure reduction in HGG patients (mean seizure number pre-PER treatment: 7.6 ± 10.5 vs. mean seizure number post: 3.4 ± 6.2; *p* = 0.01) but not in low grade glioma (LGG) patients (mean seizure number pre: 10.4 ± 16.8 vs. mean seizure number post-PER treatment: 1.4 ± 2.3; *p* = 0.10). The IDH mutate condition seemed to positively impact on the seizure outcome: IDH1 mutated patients obtained a mean number of seizure reduction from 11.4 ± 12.3 to 5.9 ± 8.8 (*p* = 0.02) while IDH non-mutated patients decreased from 11.0 ± 19.3 to 1.0 ± 1.2 (*p* = 0.13). Eleven patients (11/36; 30.6%) reported non-severe adverse events (two experienced anxiety, two irritability/aggressiveness, five dizziness, and two fatigue/tiredness). Although retention rate at 12 months was 58.3%, only three participants discontinued the medication because of the adverse events, whereas other patients dropped out because of the oncological disease progression (seven patients), non-adherence to treatment (two patients), adverse events (three patients), death (two patients), and worsening of seizures (one patient). The cognitive item at Quality of Life in Epilepsy Inventory (QOLIE) global score showed a small non-significant improvement after PER treatment (basal score: 47.0 ± 28.0; post PER: 56.9 ± 26.7; *p* = 0.19). Eleven patients (30.6%) underwent tumor progression.

Maschio et al. [15], in a prospective observational study, reported data on 26 BTRE patients (males n = 16; females n = 10; age: 32–75 years; tumor histology: eight LGG, eight HGG, seven glioblastomas, two meningiomas, one metastasis). Eleven patients were on AED monotherapy and 15 on polytherapy. The mean daily PER dosage was 6.6 mg in 21 patients who reached the final follow-up and 6.4 mg in the whole population. Responder rate at 6 months was 95.2%: seven patients were seizure-free, 13 reported a seizure reduction ≥50%, and one remained stable. Histology, IDH1-2 mutation, and mgmt methylation did not seem to influence the seizure response to PER. Four patients (15.4%) reported adverse events (vertigo: n = 4; aggressiveness: n = 1). Among them, two patients discontinued PER (one due to aggressiveness, and one due to vertigo) [15]. Nine patients underwent neuropsychological questionnaires at the baseline and at the end of the follow-up period. The authors did not find any difference in the cognitive performances.

#### 3.2.2. Studies on Glioma-Related Epilepsy

A study conducted by Maschio et al. [16] retrospectively analyzed 11 patients with glioma-related epilepsy (males n = 9; females n = 2; age: 31–76 years): two patients with anaplastic oligo-astrocytoma (AOA), two with anaplastic astrocytoma (AA), three with glioblastoma (GBM), and four with low grade astrocytoma (LGA). Five patients were on AED monotherapy and six on polytherapy. The mean daily PER dosage was 7.3 mg. After 12 months, five patients (45.5%) were seizure-free, four (36.4%) had a seizure reduction ≥50%, and two were unchanged (18.2%). The responder rate was 81.8%. Compared to the IDH1-wild type condition, the IDH1-mutation was associated with a more marked decrease in the mean number of seizures after treatment with PER (93% vs. 71.2%) and a better responder rate (100% vs. 71.4%); also, patients with methylated MGMT had a greater reduction in the mean number of seizures after treatment with PER when compared with patients without methylation. Two patients (18.2%) reported non-severe adverse events (anxiety and agitation) during treatment, which did not lead to PER discontinuation. Tumor progression during PER therapy was evidenced using brain magnetic resonance imaging (MRI) in four patients (33.6%).

Similarly, Chonan et al. [17] conducted a retrospective analysis on 11 patients with glioma and epilepsy (males n = 9; females n = 9; age: 24–76 years; tumor histology: two with diffuse astrocytoma—DA, five with AA, three with oligodendroglioma, one with anaplastic oligodendroglioma—AO, seven with GBM). All patients were treated with levetiracetam monotherapy. PER was added-on at a mean dosage of 4 mg. After a median 10.6 months of follow-up, all but one case achieved seizure freedom (responder rate: ≥94.4%). Non-severe adverse effects (irritability) occurred in two patients. These were mainly seen at the time of the initiation of therapy, disappearing after 4–6 weeks with continuous use of PER, then not leading to PER discontinuation. However, ten patients discontinued the drug as they died because of tumor progression (retention rate: 55.5%). The authors reported that PER did not increase the toxicity of radiation therapy and chemotherapy.

In a prospective study, Izumoto et al. [18] consecutively recruited 12 glioma patients (males n = 8; females n = 4; age: 31–84 years) with uncontrollable epilepsy: two patients had grade 2 glioma (1 DA and 1 oligoastrocytoma), eight patients had grade 3 glioma (three AA and five AO), and two patients had GBM. Nine patients were treated with 4 mg of PER and three patients were treated with 8 mg as maintenance doses. Ten patients achieved more than 50% seizure reduction. Among them, six patients became seizure-free. Two patients reported side effects: one had intolerable dizziness (which led to drug discontinuation), and one reported sluggish speech and dizziness after concomitant alcohol intake. Two patients discontinued the drug (one for the adverse event and one died for intratumor hemorrhage caused by head injury). The authors assessed tumor progression by analyzing the volume and peritumoral edema changes within 6 months by MRI-FLAIR images. The tumor volume decreased in eight out of nine patients over 6 months by FLAIR images and increased in one of nine patients (one patient was excluded from the analysis because of no detectable FLAIR-high lesion on the MRI). Moreover, they found a correlation between the tumor volume reduction at brain MRI (T2-FLAIR sequences) and the plasma concentrations of PER (R^2^ = 0.6909) [18].

In addition, Dunn-Pirio et al. [19] conducted a single-arm study of adjunctive PER for patients with focal-onset glioma-associated seizures. This study included eight patients (males n = 6; females n = 2) aged between 35 and 61 years (low grade astro- or oligo[astro]dendroglioma: six; anaplastic astro- or oligo[astro]dendroglioma: three; GBM: two; ganglioglioma: one). The IDH1 mutation status was available for seven patients: five were mutated while the remaining two were non-mutated. Concerning the MGMT methylation status, seven patients were methylated while in one patient, the status was unknown [19]. PER was added-on at a mean dosage of 7 mg. In this study, the most common related adverse events were fatigue and dizziness. Three out of eight participants had self-reported seizure reduction and an additional three reported improved control. Even in this case, the authors concluded that PER was safe for patients with glioma-related focal-onset epilepsy [19].

In another study, Vecht et al. [20] tested PER (mean dosage: 7.2 mg/day) in patients with drug-resistant epilepsy in LGG and HGG. Twelve patients (males n = 9; females n = 3; age: 31–65 years) were included with a 6 month median duration of follow-up. An objective seizure response (defined as 50% drop in seizure frequency or as seizure-freedom) was observed in nine (75%) out of 12 patients: 50%-seizure response in three, seizure-freedom in six. Side-effects occurred in six patients (dizziness/vertigo: four; drowsiness: two). Furthermore, cognitive function was examined by a computerized test on cognitive speed and improved in six out of the eight patients tested [20].

**Table 1 biomedicines-11-00651-t001:** Demographic and clinical data of patients receiving perampanel.

Author, Year	Study Design	Sample Size	Male/ Female	Age	Tumor Histology	IDH Mutation	mgmt Methylation	GTR N Patients	CT N Patients	Type of CT	RT N Patients	Seizure Semeiology	AED Monotherapy N Patients	AED Polytherapy N Patients	Final Dosage of Perampanel (mg/Day)	Follow-Up (Months)	N Patients at the End of Follow-Up	Primary Outcome	Secondary Outcome	Means of Cognitive Function Evaluation
Coppola, 2020	Single arm prospective	36	23/13	Median: 46 Range: 15–75	LGG: 11 HGG 14 GBM: 7 Unclassified: 4	Mutated: 6 Unmutated: 10 Unknown: 20	Unmethylated: 4 Unknown: 25	13	29	TMZ: 16 Fotoemustine: 4 Bevacizumab: 1 Other: 5 Unknown: 3	17	Focal aware: 14 Focal unaware: 7 Focal to bilateral: 11 Generalized: 4	19	17	2 mg: 1; 4 mg: 7; 6 mg: 14; 7 mg: 1; 8 mg: 9; 10 mg: 3; 12 mg: 1	12	21	Efficacy of PER (Responder rate: ≥50% reduction in seizure frequency; seizure freedom)	Retention rate; quality of life modification	QOLIE 31-P test
Maschio, 2020	Single arm prospective	26	16/10	Mean: 47.5 Range: 32–75	LGG: 8 HGG: 8 GBM: 7 Meningioma: 2 Metastasis: 1	Mutated: 6 Unmutated: 11 Unknown: 9	Unmethylated: 4 Unknown: 15	11	25	TMZ: 13 Fotoemustine: 4 Bevacizumab: 1 Other: 4 Unknown: 3	11	Focal aware: 1 Focal unaware: 6 Focal to bilateral: 7	11	15	2 mg: 1; 4 mg: 6; 6 mg: 9; 8 mg: 7; 10 mg: 2; 12 mg: 1	6	21	Efficacy of PER (Responder rate: ≥50% reduction in seizure frequency; seizure freedom)	PER-related side effects; PER impact on cognition, mood, and quality of life	Battery of cognitive tests
Maschio, 2019	Retrospective	11	9/2	Mean: 54 Range: 31–76	AOA: 2 AA: 2 GBM: 3; LGA: 4	Mutated: 3 Unmutated: 7 Unknown: 1	Unmethylated: 5 Unknown: 1	6	7	TMZ: 3 Fotoemustine: 2 Bevacizumab:1 Other:1	3	Focal: 6 Focal to bilateral: 5	5	6	6 mg: 5 8 mg: 4 10 mg: 2	12	11	Efficacy of PER (Responder rate: ≥50% reduction in seizure frequency; seizure freedom)	PER-related side effects	Not assessed
Chonan, 2020	Retrospective	18	9/9	Mean: 50 Range 24–76	DA: 2 AA: 5 OL G2: 3; AO: 1 GBM: 7	Mutated: 10 Unmutated: 5 Unknown: 3	NA	2	16	Nimustine hydrochloride (ACNU), TMZ, and bevacizumab: 15 TMZ: 1 Bevacizumab: 1 (for recurrence)	17 (1 for recurrence)	Focal: 8 Focal to bilateral: 10	18	0	2 mg: 2 4 mg: 15 8 mg: 1	1–21 months (median, 10.6 months)	18	Efficacy of PER (number of patients achieving seizure freedom)	Time to seizure freedom; number and type of adverse events	Not assessed
Izumoto, 2018	Single arm prospective	12	8/4	Mean: 57.8 Range: 31–84	DA: 1 Oligoastrocytoma: 1 AA: 3 AO: 5 GBM: 2	NA	NA	NA	9	TMZ: 6 Bevacizumab: 3	11	Focal: 7 Focal to bilateral: 5	10	2	4 mg: 9 8 mg: 3	6	12	Efficacy of PER (responder rate: ≥50% reduction in seizure frequency; seizure freedom)	tumor volume and peritumoral edema after 6 months of PER treatment	Not assessed
Dunn-Pirio, 2018	Single arm prospective	8	6/2	Median: 45 Range: 35–61	GBM: 2 AA: 2 OL: 2 DA: 2	Mutated: 5 Unmutated: 2 Unknown: 1	Unmethylated: 0 Unknown:1	NA	NA	NA	NA	Focal: 8	2	6	6 mg: 4 8 mg: 4	6	6	Seizure frequency	percentage of subjects with unacceptable adverse events	Not assessed
Vecht, 2017	Single arm prospective	12	9/3	Median: 41 Range: 31–65	Ganglioglioma: 1 Low-grade astro- or oligo(astro)dendroglioma: 6 Anaplastic astro- or oligo(astro)dendroglioma: 3 GBM: 2	NA	NA	NA	6	NA	NA	Focal aware: 7 Focal unaware: 4 Generalized: 1 Focal SE: 2.	0	12	2 mg: 1 4 mg: 2; 6 mg: 3; 8 mg: 3; 10 mg: 1; 12 mg: 2	6	12	Efficacy of PER (Responder rate: ≥50% reduction in seizure frequency; seizure freedom)	Outcome on cognition	CTCS

AEDs: antiepileptic drugs; AA: anaplastic astrocytoma; AO: anaplastic oligodendroglioma; AOA: anaplastic oligoastrocytoma; CT: chemotherapy; CTCS: computerized test for cognitive speed; DA: diffuse astrocytoma; GBM: glioblastoma; GTR: gross total resection; HGG: high grade glioma; IDH: isocitrate dehydrogenase; LGA: low grade astrocytoma; LGG: low-grade glioma; mgmt: O6-methylguanine DNA methyltransferase; OL: oligodendroglioma; NA: not available; PER: perampanel; RT: radiotherapy; SE: status epilepticus.

**Table 2 biomedicines-11-00651-t002:** Results of included studies on perampanel in patients with brain tumor-related epilepsy.

Author, Year	Seizure Freedom N Patients (%)	≥50% Re-Duction in Seizure Frequency N Patients (%)	<50% Re-Duction in Seizure Frequency N Patients (%)	Unchanged Seizure Frequency N Patients (%)	Worsening Seizure Frequency N Patients (%)	Adverse Events N Patients (%)	Type of Adverse Events	Retention Rate, N Patients (%)	Effect on Cognitive Function
Coppola, 2020	7 (33.3)	12 (57.1)	1 (4.8)	1 (4.8)	0 (0)	11 (52.4)	Anxiety: 2 Aggressiveness: 2 Dizziness: 5 Fatigue: 2	21/36 (58.3)	Improvement
Maschio, 2020	7 (33.3)	13 (61.9)	0 (0)	1 (4.8)	0 (0)	4 (19.1)	Vertigo: 4 Aggressiveness: 1	16/26 (61.5)	Stability
Maschio, 2019	5 (45.5)	4 (36.4)	0 (0)	2 (18.2)	0 (0)	2 (18.2)	Anxiety: 2 Agitation: 2	11/11 (100)	NA
Chonan, 2020	17 (94.4)	NA	NA	NA	NA	2 (11.1)	Irritability: 2	10/18 (55.6)	NA
Izumoto, 2018	6 (50)	4 (33.3)	NA	NA	0 (0)	2 (16.7)	Dizziness: 2 (one after concomitant alcohol intake)	10/12 (83.3)	NA
Dunn-Pirio, 2018	0 (0)	6 (75)	0 (0)	0 (0)	2 (25)	8 (100)	Nausea: 1 Fatigue: 5 Dizziness: 2 Somnolence: 1 Confusion: 1	6/8 (75)	NA
Vecht, 2017	6 (50)	3 (25)	0 (0)	2 (16.7)	1 (8.3)	6 (50)	Dizziness/vertigo: 4 Drowsiness: 2	10/12 (75)	Improvement: 6 Stability: 1 Worsening: 1

NA: not available.

#### 3.2.3. Overview of Registered Clinical Trials

Table 3 summarizes the registered clinical trials regarding the effect of PER on BTRE. Among them, one study has been completed and the results are discussed in the paragraph above (see Dunn-Pirio et al. [19]). Two randomized controlled trials have been withdrawn (ID: ACTRN12617000078358; NCT03636958), one of them for lack of patients. A pending non-randomized single-arm study (ID: JPRN-UMIN000026095) aims to evaluate the seizure-free rate among 20 patients with post-operative glioma receiving 8 mg/day of PER in combination with 1000 mg/day of levetiracetam for 1 year. A secondary endpoint is evaluating the progression-free survival (PFS) and overall survival (OS) of glioma patients treated with PER. Two non-randomized interventional studies are in the phase of recruitment. One (ID: NCT04497142) is a pilot study aiming to compare brain activity (high frequency oscillations in the tumor margins, measured using intraoperative electrocorticography) at the time of initial glioma resection, among participants receiving PER the day before surgery versus the standard of care treatment. The other study (ID: NCT04650204) aims to assess the efficacy of PER on the reduction in the seizure frequency (number of patients with >50% reduction in focal seizures) in patients with HGB as well as the impact of PER on OS and neuropsychological function.

## 4. Discussion

In this systematic review, seven clinical studies on PER in BTRE were analyzed. The sample size ranged from eight to 36 patients, who received add-on PER (mean dosage from 4 to 7 mg/day) for BTRE. After a 6–12 month follow-up, the responder rate (% of patients achieving seizure freedom or reduction ≥ 50% of seizure frequency) ranged from 75% to 95%, with a seizure freedom rate of up to 94%. Regarding tolerability, 11–52% of patients experienced non-severe adverse effects (most frequent: dizziness, vertigo, anxiety, irritability). The retention rate ranged from 56% to 83%. However, only up to 12.5% of patients discontinued the drug because of the adverse events. The effect of PER on cognitive function was evaluated in three studies [2,15,20], which showed an overall stability after the introduction of PER, with some data on a possible improvement [20].

Responder rate and seizure-freedom rate values were higher than those observed in randomized trials regarding the effect of PER in focal seizures [10,11,21,22] as well as in real-life studies [23]. The role of the IDH mutation in influencing the response to PER has only been evaluated in three studies, with varying results, but tending to state that IDH mutation could be a predictive factor for a good response to PER therapy. Dunn-Pirio et al. reported that most of the participants in their study who had a decrease in seizure activity had IDH1 mutant tumors. However, due to the high rate of IDH1 mutant tumors within the study population (7/8 patients), sampling error could not be excluded [19]. It is common knowledge that the R132H mutation of the IDH1 gene results in a gain of function and promotes the accumulation of D-2-hydroxyglutarate (D-2-HG), which is structurally similar to glutamate [5]. The mutation in the IDH gene is associated with an increased risk of seizures preoperatively and postoperatively, and this may be a reason for the higher incidence of epilepsy in astrocytoma and oligodendroglioma compared to GBM [5]. Moreover, PER showed a safety profile that is consistent with that demonstrated in the phase 3 studies and their extension [10,11,21,22] as well as in real-life studies [23].

In the studies analyzed, the proportion of male subjects was higher than females and the patients’ age ranged from 15 to 84 years. Recent sub analysis of phase III studies has shown a higher efficacy of PER in women [24]. This is related to gender pharmacokinetic differences that make drug clearance lower and plasma concentrations higher in females at the same dosage. Moreover, real world data confirm the efficacy and safety of PER, even in the elderly [25].

Regarding cognitive functions, the methods used to assess cognitive domains were variable between studies, and only one used multiple standardized cognitive tests [15]. However, these results are in line with previous reports regarding the impact of PER on the objective cognitive measures in patients with epilepsy [26].

Regarding the possible anti-tumor effect of PER, the studies currently carried out do not report data on parameters such as OS and PFS. Four studies reported data about the number of patients undergoing progression during PER therapy [2,15,16,17]. Izumoto et al. assessed tumor progression by analyzing the volume and peritumoral edema changes within 6 months by MRI-FLAIR images. They found a correlation between the tumor volume reduction and the plasma concentrations of PER. However, data on the effect of PER on contrast enhancing portions of the tumor were not reported, so no clear distinction between the effective tumor volume and peritumoral edema/inflammation was made. Among the ongoing trials, two non-randomized studies (ID: NCT04650204; JPRN-UMIN000026095) have reported the evaluation of PFS and/or OS among glioma patients as a secondary objective.

Although informative, the studies conducted so far have some limitations including the low number of patients recruited, the single arm design, and the recruitment of heterogeneous patients by histology, tumor grade, and concomitant antitumor and antiepileptic treatment. Moreover, with the advent of the new WHO classification of brain tumors in 2021 [27], some different approaches to brain tumor nomenclature and grading have changed, and new molecular entities have been defined (e.g., IDH-mutant grade 4 astrocytoma and IDH-wild type molecularly defined glioblastoma), whose behavior in terms of prognosis and response to standardized treatment remains to be defined. Therefore, future randomized-controlled studies addressing specific molecular subtypes are needed.

## 5. Conclusions

PER seems efficacious and safe in the treatment of BTRE. However, the data currently supporting the use of PER in the treatment of BTRE are hampered by several limitations. Further studies should be conducted in more homogeneous and larger populations, testing for efficacy, safety, and tolerability. Given the potential cytostatic role of PER shown in preclinical in vitro studies, it would be interesting to include evaluations of the effect of PER on tumor progression, OS, and PFS in future trials.

## Figures and Tables

**Figure 1 biomedicines-11-00651-f001:**
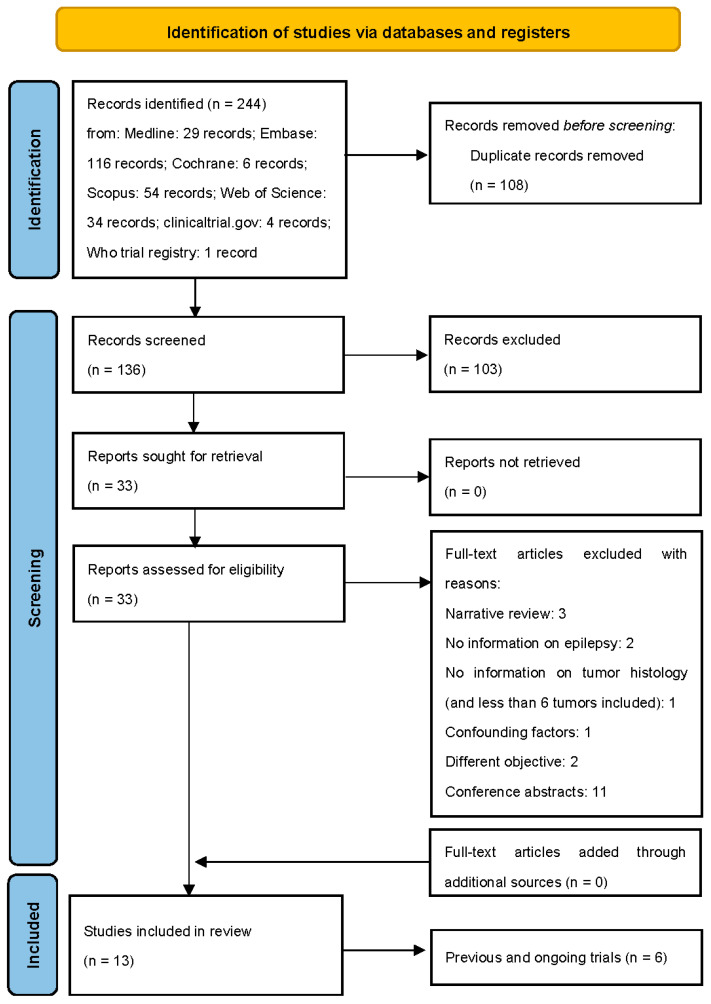
Flow diagram of the study selection process.

**Table 3 biomedicines-11-00651-t003:** Clinical trials assessing efficacy of perampanel in brain tumor-related epilepsy (BTRE).

Name (ID) (Year)	Study Design	Status	Population	Intervention	Comparison	Outcome
NCT02363933 (2015)	Phase 2 single-arm study	Completed (see Dunn-Pirio et al.) [19]	9 patients diagnosed with glioma and refractory partial onset seizure activity (defined as 3 or more seizures in a 28-day period) on levetiracetam monotherapy	Perampanel + current anti-epileptic drug	Current anti-epileptic drug	Percentage of patients with ≥50% seizure reduction during the maintenance period compared with seizure frequency before initiation of perampanel
JPRN-UMIN000026095 (2017)	Single arm non-randomized	Pending	20 post-operative glioma patients with epilepsy	Addition of 8 mg of perampanel on the patients treated with 1000 mg of levetiracetam for 1 year	NA (single arm non-randomized)	Seizure free rate (Time Frame: 12 months after treatment); Overall survival; Progression-free survival
ACTRN12617000078358 (2017)	Randomized controlled trial	Stopped early	40 patients with radiological diagnosis of a supratentorial WHO grade II-III glioma who experienced a pre-operative seizure attributed to glioma	Perampanel increased to 6 mg/d	Levetiracetam	Proportion of patients seizure-free for 24 or more continuous weeks in assessment phase (weeks 5–52)—assessed by seizure diary; Time to first seizure in assessment phase (weeks 5–52)—assessed by seizure diary
NCT04497142 (2020)	Non-randomized interventional	Recruiting	20 patients with radiologic evidence of anaplastic astrocytoma or GBM.	Predetermined first dose of perampanel on the day before their tumor surgery	Standard of care	Rate of high frequency oscillations [Time Frame: Peri-operative]
NCT04650204 (2020)	Non-randomized interventional	Recruiting	40 patients with a diagnosis of biopsy-proven high-grade glioma and epilepsy refractory to at least 1, drug.	Perampanel	Conventional antiepileptic treatment	Number of patients with a high-grade glioma who achieve a >50% reduction in focal seizures with perampanel after failing 1 or more anti-seizure drugs at 3 and 6 months; overall survival; decline in neuropsychological function.
NCT03636958 (2021)	Randomized controlled trial	Withdrawn (concomitant decision of the sponsor and the PI, lack of patients)	Patients with a diagnosis of glioma-refractory epilepsy	Perampanel	Conventional antiepileptic treatment	Monthly frequency of seizures

NA: not available.

## Data Availability

Not applicable.

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
