# Peer review of "Efficacy and Tolerability of Perampanel in Brain Tumor-Related Epilepsy: A Systematic Review"

_biomedicines, 2023, doi:10.3390/biomedicines11030651_

Round 1

Reviewer 1 Report

In this systematic review, the literature on efficacy and tolerance of the selective non-competitive α‑amino‑3‑hydroxy‑5‑methyl‑4‑isoxazolepropionic acid receptor antagonist Perampanel PER in brain tumor-related epilepsy (BTRE) was reviewed. It was concluded that PER seems efficacious and safe in the treatment of BTRE.

The following phrases concerned me and can require correction.

1.       I understand that the size of Abstract is too small to include all important results of this review. However, the result subsection seems to be very small (Results: seven studies and six clinical trials were included. Responder rate ranged from 75% to 95%. Non-severe adverse effects were found in 11-52% of patients. Retention rate ranged from 56% to 83%.” Why don’t the authors increase it at expense of other subsections, especially Conclusion (see also 10).

2.       “The search was then restricted to studies with full text available in English language.” Does it simply mean that the papers in other than English languages were excluded?

3.       “Whenever the retrieved information was unclear or there was uncertainty about the inclusion of one study, the decision was taken after a consensus-based discussion.” The description of this consensus based decision-making process can be reworded. Moreover. “A third author (F.V.) supervised the entire systematic review process”. Has this F.V. influence on this consensus based decision-making process or the function of F.V. was limited to just overseeing or watching over the process?

4.       “Among these, were duplicates and then removed.” In Figure 1. Information is lost in one of the squares of “Identification” (“Records identified (n = 207) from: Medline: 25 records; Embase: 85 records; Cochrane: 6 records; Scopus: 54 records; Web of”).

5.       It is difficult to understand the difference between Table1 and Table S1 (the last table was also included in the article text on Page 14). Why don’t the authors combine them in a single table of Supplementary?

6.       How about the moving of Table 2 in the supplementary file?

7.       Can Table 3 be improved for its better reading? For instance, the 1st column can become wider after sending one line (“Type of adverse events”) to the table notes. The same can be done for the last cell of this table.

8.       For its better reading, the first columns of Table 4 can be merged.

9.       The first paragraph of Discussion must contain the brief description of the review results, while the comparison with other publications can be given in the next paragraphs.

10.   “5. Conclusions”. This section mostly reproduces the conclusion subsection of Abstract.

Author Response

RESPONSE TO REVIEWER #1

  1. I understand that the size of Abstract is too small to include all important results of this review. However, the result subsection seems to be very small (Results: seven studies and six clinical trials were included. Responder rate ranged from 75% to 95%. Non-severe adverse effects were found in 11-52% of patients. Retention rate ranged from 56% to 83%.” Why don’t the authors increase it at expense of other subsections, especially Conclusion (see also 10). We thank the reviewer for this suggestion. Accordingly, we modified the abstract by detailing the results section.
  1. “The search was then restricted to studies with full text available in English language.” Does it simply mean that the papers in other than English languages were excluded? We thank the reviewer for pointing out this issue; we confirm that papers in other than English languages were excluded. We added this information in the Methods section.
  2. “Whenever the retrieved information was unclear or there was uncertainty about the inclusion of one study, the decision was taken after a consensus-based discussion.” The description of this consensus based decision-making process can be reworded. Moreover. “A third author (F.V.) supervised the entire systematic review process”. Has this F.V. influence on this consensus based decision-making process or the function of F.V. was limited to just overseeing or watching over the process? Response: we thank the reviewer for the comment. We modified the sentence as follows: “Any disagreement between the two investigators on the inclusion of a study was solved by involving a third investigator (F.V.) in the discussion to reach a consensus decision.”
  1. Among these, were duplicates and then removed.” In Figure 1. Information is lost in one of the squares of “Identification” (“Records identified (n = 207) from: Medline: 25 records; Embase: 85 records; Cochrane: 6 records; Scopus: 54 records; Web of”). Response: we thank the reviewer for the comment. We apologize for the wrong size of the box which hampered the visibility of our report. We resized the box and the commented issue is fixed.
  2. It is difficult to understand the difference between Table1 and Table S1 (the last table was also included in the article text on Page 14). Why don’t the authors combine them in a single table of Supplementary? We thank the reviewer for this suggestion; we combined Table 1 and Table S1 in just one table (Table S1).
  3. How about the moving of Table 2 in the supplementary file? We thank the reviewer for the suggestion. However, we would prefer to keep it in the main text because the Academic Editor asked to provide the information on age and gender distribution of patients.
  4. Can Table 3 be improved for its better reading? For instance, the 1stcolumn can become wider after sending one line (“Type of adverse events”) to the table notes. The same can be done for the last cell of this table. We thank the reviewer for the suggestion that, along with the suggestion of Reviewer #2, was addressed by modifying the tables in order to improve the ease of reading.
  5. For its better reading, the first columns of Table 4 can be merged. Also this suggestion was implemented.
  6. The first paragraph of Discussion must contain the brief description of the review results, while the comparison with other publications can be given in the next paragraphs. Response: We thank the reviewer for the suggestion. We reported a summary of the review results in the first paragraph and discussed the other publications in the following paragraphs, as suggested.
  7. “5. Conclusions”. This section mostly reproduces the conclusion subsection of Abstract. Response: We thank the reviewer for the comment. We have modified the abstract by increasing the results section, which forced us to reduce the size of the conclusion section.

Reviewer 2 Report

Introduction: Might discuss more extensively why this systematic review is warranted. Have previous systematic reviews been conducted evaluating the efficacy of antiseizure medications in brain tumor patients? If so, what kind of limitations did these have, which warrant a systematic review specifically assessing the efficacy of PER in BTRE. Please discuss more about different brain tumors in the introduction. 

Methods: The search strategy is seriously limited. Please consult a librarian who can help in setting up a proper search strategy with sufficient synonym keywords (e.g., meningioma, brain metastasis, seizure etc.). Not using a more comprehensive search strategy might have resulted in only identifying the best known studies assessing the efficacy of perampanel. In addition, please extend the inclusion criteria. Would for example studies assessing primary prophylaxis of perampanel be included?

Results: Table 2 and 3 are difficult to comprehend due to an uncommon choice. Please change direction of the tables with manuscripts on the y-axis and study characteristics on the x-axis. 

Do not use uncommon abbreviations such as GRE, but just write the whole term.

Conclusion: Authors discuss dat for use of PER in BTRE is hampered by several confounding factors. However, they do not discuss which confounding factors they mean in the discussion section. Please dive more deeply into this methodological discussion.  

Author Response

RESPONSE TO REVIEWER #2

  1. Introduction: Might discuss more extensively why this systematic review is warranted. Have previous systematic reviews been conducted evaluating the efficacy of antiseizure medications in brain tumor patients? If so, what kind of limitations did these have, which warrant a systematic review specifically assessing the efficacy of PER in BTRE. Please discuss more about different brain tumors in the introduction. Response: We thank the reviewer for this suggestion: the importance of focusing on Perampanel in this review has been discussed in the introduction, as suggested.  We have also added the following sentence: “The role of glutamate in brain tumor-related epileptogenesis is widely known for gliomas, but have been established in other histotypes, including meningiomas and brain metastases”

  1. Methods: The search strategy is seriously limited. Please consult a librarian who can help in setting up a proper search strategy with sufficient synonym keywords (e.g., meningioma, brain metastasis, seizure etc.). Not using a more comprehensive search strategy might have resulted in only identifying the best-known studies assessing the efficacy of perampanel. In addition, please extend the inclusion criteria. Would for example studies assessing primary prophylaxis of perampanel be included? We thank the reviewer for pointing out this important issue. As suggested, we have extended the search strategy, including more keywords in order to enlarge the literature search. Furthermore, we tried to find studies by assessing the primary prophylaxis of perampanel, but no studies have been published so far. We also chose to include studies in which data on seizures and tumor histology could be available in a sufficient number of patients (n ≥ 5 patients).

  1. Results: Table 2 and 3 are difficult to comprehend due to an uncommon choice. Please change direction of the tables with manuscripts on the y-axis and study characteristics on the x-axis.  We thank the reviewer for pointing out this issue as suggested we have changed the directions of tables 2 and 3 to improve the ease of reading consultation of the tables.

  1. Do not use uncommon abbreviations such as GRE, but just write the whole term. Response: We thank the reviewer for the suggestion: we have removed the abbreviation “GRE” from the manuscript.

  1. Conclusion: Authors discuss data for use of PER in BTRE is hampered by several confounding factors. However, they do not discuss which confounding factors they mean in the discussion section. Please dive more deeply into this methodological discussion.  Response: we thank the reviewer for pointing out this issue and we totally agree. We have substituted the term “confounding factors”, with “limitations”, referring to the limitations we have considered in the “discussion” section.

Round 2

Reviewer 2 Report

No further comments